# CuMo_x_W_(1-x)_O_4_ Solid Solution Display Visible Light Photoreduction of CO_2_ to CH_3_OH Coupling with Oxidation of Amine to Imine

**DOI:** 10.3390/nano10071303

**Published:** 2020-07-03

**Authors:** Chao Luo, Tian Yang, Qianfei Huang, Xian Liu, Huan Ling, Yuxin Zhu, Guoming Xia, Wennan Zou, Hongming Wang

**Affiliations:** 1Institute for Advanced Study, Nanchang University, Nanchang 330031, China; luochao@ncu.edu.cn (C.L.); huanling_ncu@126.com (H.L.); ncu_zhuyuxin@163.com (Y.Z.); gmxia_ncu@126.com (G.X.); 2College of Chemistry, Nanchang University, Nanchang 330031, China; yangtian_92@163.com (T.Y.); lzhnxydlx@163.com (X.L.); 3School of Information Engineering, Jiangxi Modern Polytechnic College, Nanchang 330095, China; huangqianfei1015@hotmail.com

**Keywords:** photocatalysis, visible light, solid solution, reduction of CO_2_

## Abstract

The photoreduction of carbon dioxide (CO_2_) to valuable fuels is a promising strategy for the prevention of rising atmospheric levels of CO_2_ and the depletion of fossil fuel reserves. However, most reported photocatalysts are only active in the ultraviolet region, which necessitates co-catalysts and sacrificial agents in the reaction systems, leading to an unsatisfied economy of the process in energy and atoms. In this research, a CuMo_x_W_(1-x)_O_4_ solid solution was synthesized, characterized, and tested for the photocatalytic reduction of CO_2_ in the presence of amines. The results revealed that the yield of CH_3_OH from CO_2_ was 1017.7 μmol/g under 24 h visible light irradiation using CuW_0.7_Mo_0.3_O_4_ (x = 0.7) as the catalyst. This was associated with the maximum conversion (82.1%) of benzylamine to *N*-benzylidene benzylamine with high selectivity (>99%). These results give new insight into the photocatalytic reduction of CO_2_ for valuable chemical products in an economic way.

## 1. Introduction

For energy conservation and environmental protection, direct conversion of CO_2_ into a source of carbon fuels is an ideal solution [1,2,3,4,5,6,7]. Among the different technologies, solar photocatalytic conversion of CO_2_ is the most promising because it only needs sunlight at room temperature and ambient pressure, which is clean, safe, and abundant [8,9,10,11,12,13,14]. As such, the development of an efficient photocatalyst under sunlight irradiation is attractive and has become a research hotspot. Over past decades, various photocatalytic materials have been analyzed: TiO_2_ [15,16,17], CdS [18,19], Bi_2_O_3_ [20,21], CeO_2_ [22,23], and Bi_2_WO_6_ [24,25,26]. Their well-designed heterostructures have been examined for reduction of CO_2_ to fuels, such as CO [27,28,29], CH_4_ [30,31,32], CH_3_OH [33,34], and C_2_H_6_ [35,36], but the overall photoconversion efficiency and product selectivity is still unsatisfactory and requires further promotion for practical applications. 

It has been confirmed that forming a solid solution between semiconductors is an excellent method in the development of visible light-driven photocatalysts for sensitive photoreduction of CO_2_ [37,38,39,40] and is widely applied in photocatalytic water splitting and pollutant degradation, while exhibiting better performance than single components comprising the solid solution [41,42,43,44,45,46]. Taking this into account, a series of solid solution photocatalysts, such as BiOBr_x_Cl_1-x_ [47], Zn_x_Cd_1-x_S [48], GaN:ZnO [49], and zinc gallogermanate [39], have been synthesized and examined for CO_2_ reduction under visible light irradiations. Very recently, Ling and colleagues [50] demonstrated that Cu_x_Ag_y_In_z_Zn_k_S solid solutions customized with RuO_2_ or a Rh_1.32_Cr_0.66_O_3_ co-catalyst showed high photocatalytic activity for the reduction of CO_2_ into CH_3_OH with a yield up to 118.5 μmol g^–1^ h^–1^ under visible light irradiation. Liang’s group [51] reported that a N-doped, graphene-functionalized Zn_1.231_Ge_0.689_N_1.218_O_0.782_ solid solution exhibited high photocatalytic activity for the evolution of CH_4_ from the photocatalytic reaction of CO_2_ and H_2_O, coupled with oxidation of benzyl alcohol under visible light irradiation.

Since the pioneering work of Benko, CuWO_4_ has been confirmed as a promising photoanode material for photoelectrochemical (PEC) splitting water, with a narrower band gap of 2.3 eV that allows for the use of visible light and enhanced stability in neutral and moderate to basic pH [52,53]. In this work, a CuMo_x_W_(1-x)_O_4_ solid solution was synthesized by a facile hydrothermal method and then examined for the photocatalytic reduction of CO_2_ under visible light irradiation without the addition of any co-catalysts and sacrificed reagents in the system. In this process, CO_2_ was reduced to CH_3_OH by photo-induced electrons, while amine was selected as the “hydrogen-donor” to capture the photogenerated holes. Then, the amine was converted into imine, resulting in good energy and atomic economies. 

## 2. Experimental 

### 2.1. General

All chemicals and solvents were purchased from commercial suppliers and used as received unless explicitly stated. Using tetramethylsilane (TMS) as an internal standard, ^1^H nuclear magnetic resonance (NMR) and ^13^C NMR spectra were measured on a Bruker AVANCE 400 spectrometer (Basel, Switzerland) in CDCl_3_. X-ray diffraction (XRD) data were collected on a D8 diffractometer from Bruker Instruments (Basel, Switzerland), utilizing Cu K_α_ radiation at a scan rate of 0.05 s^–1^. The scanning electron microscopy (SEM) images were obtained by a Quanta 200F FEI (Los Angeles, USA). Transmission electron microscopy (TEM) analysis was carried out on a JEM-2100 (TEM) (Okinawa, Japan), and energy dispersive X-ray spectra (EDX) were obtained at an accelerating voltage of 200 kV. X-ray photoelectron spectroscopy (XPS) data were performed with a Thermo Scientific (New York, USA) ESCALAB250Xi XPS spectrometer with Al Kα at 500 eV. The ultraviolet-visible (UV-Vis) DRS spectra of photocatalysts were conducted using a Cary 60 UV-vis spectrophotometer (London, Europe) with BaSO_4_ as a reference. Photoluminescence (PL) characterizations were carried out on a FluoroMax-4 spectrometer (London, Europe), using a λ_ex_ = 320 nm light source. Gas chromatography (GC) product analysis was performed on a flame ionization detector. The composition of liquid products was analyzed by GC-MS (Beijing, China) with a HP-5 capillary column (30 m × 0.25 mm × 0.25 mm).

### 2.2. The Preparation of Solid Solution Photocatalysts

A series of CuMo_x_W_(1-x)_O_4_ solid solution photocatalysts was prepared by a typical hydrothermal synthesis method, in which X is defined as the molar ratio of Mo/(Mo + W). Setting the preparation of CuMo_0.1_W_0.9_O_4_ (X = 0.1) as an example: 10 mmol Cu(NO_3_)_2_·3H_2_O was dissolved in 250 mL distilled water as precursor solution A. Onethousandth of a mol of NaMoO_4_·2H_2_O and 9.0 mmol NaWO_4_·2H_2_O were dissolved in 250 mL distilled water as precursor solution B. Precursor solution A was added dropwise to precursor solution B under a strong stirring condition in 30 min, during which the mixtures turned blue-green and more suspended substance appeared. After the addition, the suspended substance (about 80 mL) was then transferred into a 100 mL autoclave and heated at 180 °C for 24 h. After cooling to room temperature, the obtained product was then filtered and washed with distilled water and anhydrous alcohol several times and then dried in a vacuum at 80 °C for 10 h. The other photocatalysts could be easily prepared in the same way by adjusting the molar ratio: Mo/(Mo + W).

### 2.3. Photocatalytic Reduction of CO_2_

Photocatalytic reactions were carried out in 40 mL anhydrous acetonitrile with 0.06 g corresponding to the photocatalyst and 1 mmol benzylamine (purified by distilled before using) in an 80 mL self-made quartz reactor at 0.5 MPa CO_2_ partial pressure. Before the reaction, the reactor was evacuated with high-purity dry CO_2_ gas and blown through several times to remove lesser quantities of oxygen, and then it was tightly closed. Before illumination, the reaction system was stirred under darkness to prompt CO_2_ adsorption and desorption equilibrium after blowing CO_2_ for 30 min. Then, a Xe lamp (300 W) with a 420 nm cut-off filter was turned on as the light source for the photocatalytic reactions. During the processes, the reaction temperature was maintained at room temperature with a circulating cool water bath. After the reactions were complete, the products were immediately analyzed by GC (Appendix A), ^1^H NMR (Appendix A), and MS (Appendix A). Detailed calculations of methanol yield, conversion rate of benzylamine, and selectivity of N-benzylidenebenzylamine are shown in the Supporting Information. 

## 3. Results and Discussion

### 3.1. Catalyst Characterization

#### 3.1.1. XRD and EDX Analysis

To reveal the phase structure of various photocatalysts prepared by the hydrothermal method, XRD patterns were recorded and shown in Figure 1. The CuMo_x_W_(1-x)_O_4_ photocatalysts were formed in two different crystalline structures, depending on their chemical composition. For x ≤ 0.7, the XRD patterns exhibit several broad diffraction peaks with low intensity, suggesting that these photocatalysts were in a poor crystalline state. Moreover, the patterns of these materials are in good agreement with the standard spectrum of CuWO_4_·H_2_O [54,55] (JCPDS no.: 33-0503). For x ≥ 0.8, several sharp diffraction peaks appeared and intensified, indicating that these photocatalysts are in a good crystalline state. 2θ values of 20.4, 21.4, 24.9, 25.3, 25.4, and 25.9 correspond to the Cu_3_(MoO_4_)_2_(OH)_2_ of the (021) (101), (121), (130), (040), and (111) faces, respectively, which is in agreement with the reported values [56]. It can be interpreted that the increased Mo content had some effects on the crystal structure of these photocatalysts in the system [56]. Furthermore, the EDX analysis results of the sample are shown in Figure 2 and Table 1. The Mo content in the reaction system gradually increased with increasing x values, and the sample changed from CuWO_4_·H_2_O to Cu_3_(MoO_4_)_2_(OH)_2_. In addition, the XPS spectra of the samples were obtained (Appendix A), which clearly indicate that the Cu and W atoms exist in the form of Cu^2+^ and W^6+^, respectively. 

#### 3.1.2. SEM and TEM Analysis

Figure 3 shows the SEM and TEM micromorphology of the prepared samples. As seen in Figure 3a, the surface of the selected material prepared by the hydrothermal method was smooth and existed in globular particles. To observe the microstructure of the photocatalyst, the characterization of the photocatalyst (x = 0.7) was carried out using TEM, and the average particle size was shown to be about 30 nm (Figure 3b). The high resolution transmission electron microscopy (HR-TEM) images of the interfacial structure of the photocatalyst in different regions is shown in Figure 3c,d, indicating that the photocatalyst CuMo_0.7_W_0.3_O_4_ (x = 0.7) prepared via hydrothermal method was amorphous because no obvious lattice stripes were observed in the photocatalyst.

#### 3.1.3. UV-Vis Absorption and PL Spectroscopy Analysis

UV-vis DRS spectra of the solid solutions were obtained, and the results are shown in Figure 4. The samples had a wide absorption with a wavelength range from 350 nm to 750 nm. A bathochromic shift absorption band (from 570 to 610 nm) was observed with increasing x values from 0 to 0.7. With a further increasing x value (from 0.8 to 1.0), a blue-shifted absorption band was observed from 620 nm to 580 nm. The different trends of the absorption bands can be verified via XRD spectra. When x ≤ 0.7, these samples were in the form of CuWO_4_ ·H_2_O, but, when x > 0.7, they were in the form of ${\mathrm{Cu}}_{3}{{(\mathrm{MoO}}_{4})}_{2}{\left(\mathrm{OH}\right)}_{2}$. Generally speaking, the absorption of the lower band is more easily disturbed by the introduction of Mo and W than the higher bands, which indicates that a charge transfer from $O^{2-}$ to $Mo^{6+}$ or $W^{6+}$ contributes to the formation of the absorption band. This is consistent with a paper by Gaudon et al. [57,58].

As is well known, photoluminescence (PL) emission can be used to discover carrier transfer and recombination efficiency because it is caused by the recombination of photogenic carriers. Lower PL emission intensity suggests a lower recombination rate and exhibits higher photocatalytic activity for a photocatalyst [59,60], as shown in Figure 5, as the value of x increases from 0 to 0.7, the fluorescence intensity of the samples gradually weaken. This is mainly because the Mo content is gradually increasing, which inhibits the recombination of photogenerated charges and carriers, thereby improving the photocatalytic activity. The fluorescence intensity of the samples was gradually strengthened with further increases in x value (from 0.8 to 1.0). It is obvious that the PL emission spectra of samples can be divided into two categories (Figure 5). One type of spectrum was obtained for x values in the range of 0.8 to 1.0, showing strong PL emission with emission intensity increasing with increasing x value. Another emission intensity, lower than the former, was found when the x values were between 0 to 0.7. All optical results revealed that the photocatalytic activity of CuMo_0.7_W_0.3_O_4_ (x = 0.7) is optimal for the system.

### 3.2. Photocatalytic Reduction of Carbon Dioxide Activity Measurement

To investigate the performance of various photocatalysts prepared hydrothermally, the photocatalytic oxidation of amine to imine coupling with the reduction of CO_2_ to CH_3_OH was carried out under visible light (λ > 420 nm) irradiation and a 0.5 MPa CO_2_ pressure environment. The detailed results are shown in Table 2. These results prove that CH_3_OH was selectively obtained from CO_2_, and imine was produced from the oxidation of amine, which were further confirmed with GC, NMR, and MS analysis (Appendix A). We found that benzylamine can be converted to *N*-benzylidene benzylamine with high selectivity (>99%) and CO_2_ can be converted to CH_3_OH with good productivity after 12 h irradiation with photocatalysts. Moreover, the results suggested that CuW_0.7_Mo_0.3_O_4_ (x = 0.7) was the optimum photocatalyst among the examined samples, which showed the highest conversion of benzylamine (57.2%) and high selectivity towards N-benzylidenebenzylamine (>99%) within 14 h irradiations (Table 2, entry 4), and received the highest productivity of CH_3_OH up to 671.8 μmol/g. These results are almost consistent with optical observations. Therefore, the CuW_0.7_Mo_0.3_O_4_ (x = 0.7) was selected as the photocatalyst in the following experiments. In addition, the results of two blank reactions showed that the conversion of benzylamine was 7.1 and 91.3% in the absence of catalyst and in the presence of catalyst, respectively.

The time plot for the oxidation of benzylamine to the corresponding imine under 0.5 MPa of CO_2_ on CuW_0.7_Mo_0.3_O_4_ (x = 0.7) is shown in Figure 6a,b. With prolonged irradiation time, the conversion of benzylamine was significantly enhanced, accompanied by an increased yield in CH_3_OH. As shown, the yield of CH_3_OH increased slightly within 10 h, and then obviously increased as time continued. With visible irradiation for 24 h, 82.1% of benzylamine was converted and >99% of them were selectively transformed to the corresponding imine, and the CO_2_ was converted to CH_3_OH with high productivity up to 1017.7 μmol/g, suggesting a satisfactory photocatalytic efficiency and selectivity under visible light conditions.

Motivated by preceding results, the substrate scope of the oxidation of other amines to corresponding imines coupling with the reduction of CO_2_ to CH_3_OH with photocatalytic CuW_0.7_Mo_0.3_O_4_ (x = 0.7) was also carried out. Detailed results, shown in Table 3, show that the oxidative coupling reactions of the amine derivatives to their corresponding imines coupling with the formation of CH_3_OH from CO_2_ could proceed well under the same conditions. However, different substrates can present great differences in the conversion of amines and the yield of CH_3_OH. Methyl-substituted benzylamines at the *o*-, *m*-, and *p*- positions of the benzene ring (Table 3, entries 2–4) can also be converted, but there are slight differences in the conversion process, and it can been seen from the Table 3 that their conversion rates differ vary greatly, which might to be relative to the effect of the steric hindrance. It can be ascribed to the electronic effects associated with electron withdrawing substituents (entry 5–6) and electron donating substituents (entry 4,7) on the benzene ring have an obvious effect on the conversion rate of amines, as well as the yield of formation of CH_3_OH. Two halo-substituted benzylamines could be oxidized to corresponding imines with a conversion of 30% and yield of 221.4 μmol/g of CH_3_OH. Moreover, heterocyclic amines were also transformed into the corresponding imines (entries 9–10). However, when aliphatic amines were selected as the substrates (Table 3, entry 11,12), no imines were obtained.

To measure the reusability of the photocatalyst, the CuW_0.7_Mo_0.3_O_4_ (x = 0.7) was selected and re-used with the same reaction conditions three times. Before the proceeding reaction, the used photocatalysts were purified by thorough washing with CH_3_CN. Figure 7 illustrates cycling runs of CuW_0.7_Mo_0.3_O_4_ (x = 0.7) in the photocatalytic reduction of CO_2_ coupling with amine selective oxidation to imine under visible light irradiation. It is clear that the high photocatalytic activity of the as-prepared sample minimally decreased after three recycles, indicating good photocatalytic stability of the photocatalyst under visible light irradiation. The decrease in photocatalytic efficiency was mainly due to the loss of crystal water in the photocatalyst after use.

The mechanism for photocatalytic reduction of CO_2_ to CH_3_OH coupling with the conversion of amine to corresponding imine is similar to our previous report [61] as shown in Figure 4. Details are as follows:Catalyst + hv → catalyst^∗^ + e^-^_cond_ + p^+^_val_,(1)
PhCH_2_NH_2_ + 2p^+^_val_ → PhCH=NH + 2H^+^_cond_,(2)
CO_2_ + 2H^+^_cond_ + 2e^-^_cond_ → HCOOH,(3)
HCOOH + 2H^+^_cond_ + 2e^-^_cond_ → HCHO + H_2_O,(4)
HCHO + 2H^+^_cond_ + 2e^-^_cond_ → CH_3_OH,(5)
PhCH=NH + H_2_O → PhCHO + NH_3(S)_,(6)
PhCHO + PhCH_2_NH_2_ → PhCH=NCH_2_Ph + H_2_O,(7)

With absorption of visible light (Equations (1) and (2)), the photogenerated holes in the valence band (VB) of the photocatalyst oxidized benzylamine to the corresponding imine and generated H^+^ (Equation (2)), which shows that CO_2_ can be reduced by photoelectrons in conduction band (CB) to generate HCOOH, HCHO, and CH_3_OH (Equations (3)–(5)). The formed imine in Equation (2) was easily decomposed into benzaldehyde and NH_3(S)_ (Equation (6)), and then the obtained benzaldehyde directly reacted with the benzylamine to form the corresponding imine and H_2_O (Equation (7)).

## 4. Conclusions

In summary, a series of CuW_x_Mo_(1-x)_O_4_ solid solutions were prepared and applied to the reduction of CO_2_ into CH_3_OH coupling with the conversion of aromatic amine into its corresponding imine. The results revealed that the yield of CH_3_OH from CO_2_ was 1017.7 μmol/g under 24 h visible light irradiations using an optimized photocatalyst, CuW_0.7_Mo_0.3_O_4_ (x = 0.7). The catalyst chosen was associated with the maximum conversion (82.1%) of benzylamine to N-benzylidene benzylamine with high selectivity (>99%). Further experiments revealed that CuW_0.7_Mo_0.3_O_4_ (x = 0.7) exhibited good substrate suitability, photocatalytic activity, and photocatalytic stability. All results should be helpful for the design and application of economical photocatalysts for the reduction of CO_2_ to CH_3_OH under visible light.

## Figures and Tables

**Figure 1 nanomaterials-10-01303-f001:**
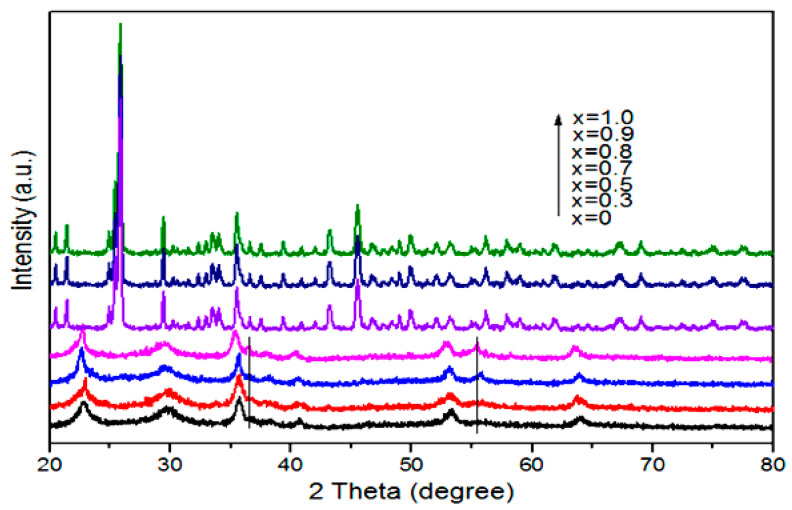
XRD patterns of nanoparticles prepared by hydrothermal method at 180 °C.

**Figure 2 nanomaterials-10-01303-f002:**
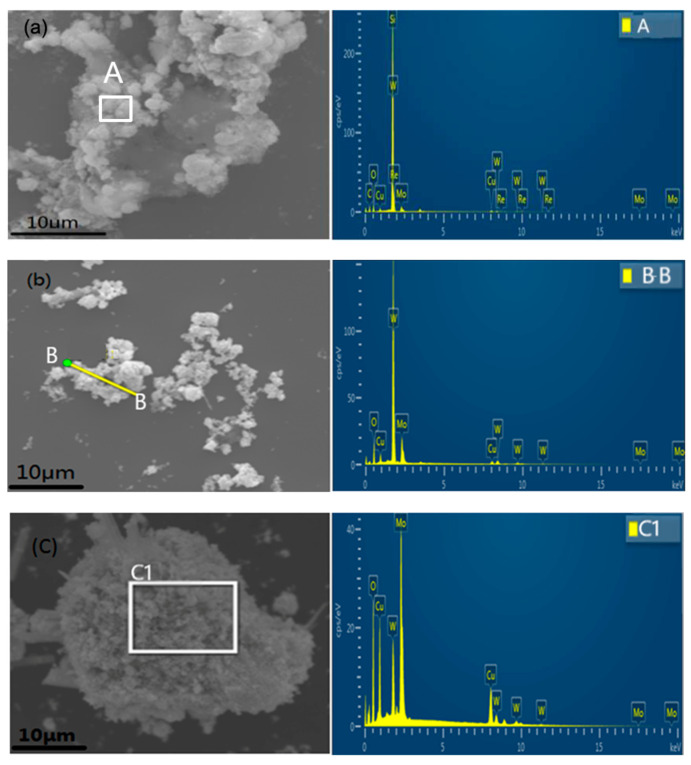
Energy dispersive X-ray spectra (EDX) spectra of (**a**) CuMo_0.5_W_0.5_O_4_ (x = 0.5) and the plane (**A**), (**b**) CuMo_0.7_W_0.3_O_4_ (x = 0.7) and the line (**B-B**), and (**c**) CuMo_0.8_W_0.2_O_4_ (x = 0.8) and the plane (**C1**).

**Figure 3 nanomaterials-10-01303-f003:**
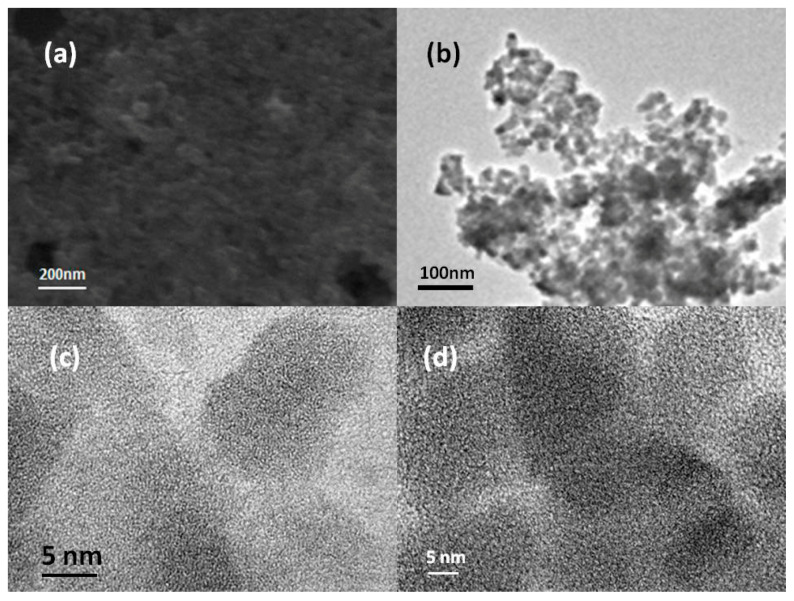
(**a**) Scanning electron microscopy (SEM), (**b**) transmission electron microscopy (TEM), and (**c**), (**d**) high resolution transmission electron microscopy (HR-TEM) images of the CuMo_0.7_W_0.3_O_4_ (x = 0.7) photocatalyst.

**Figure 4 nanomaterials-10-01303-f004:**
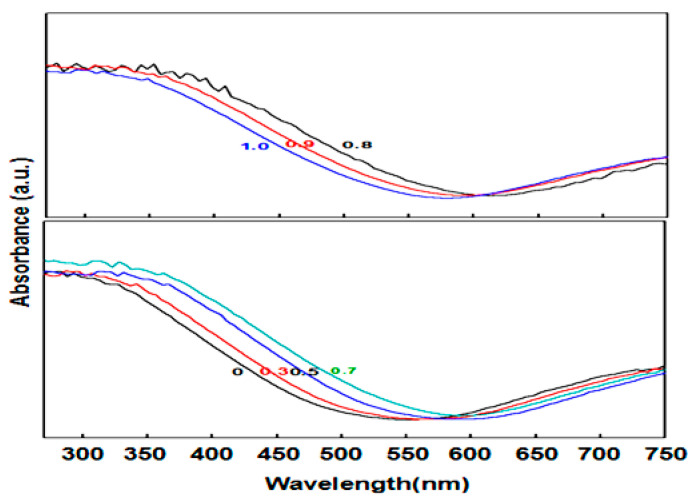
Ultraviolet-visible (UV-vis) diffuse reflectance spectra of different samples.

**Figure 5 nanomaterials-10-01303-f005:**
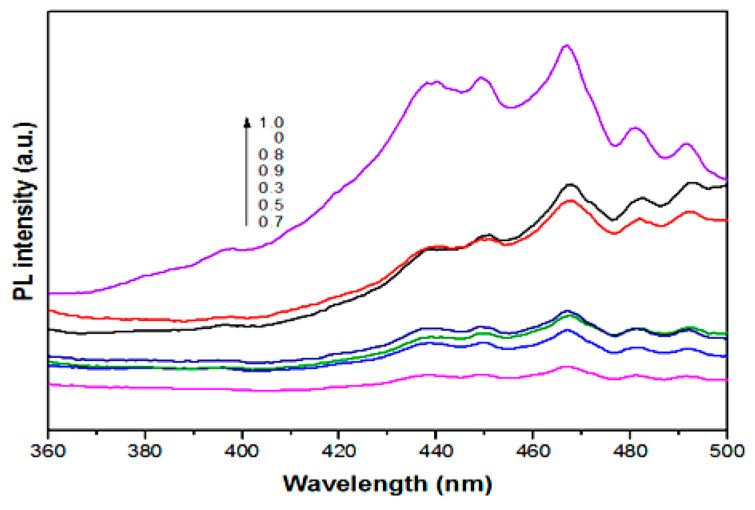
Photoluminescence emission spectra of different samples with excitation at 320 nm.

**Figure 6 nanomaterials-10-01303-f006:**
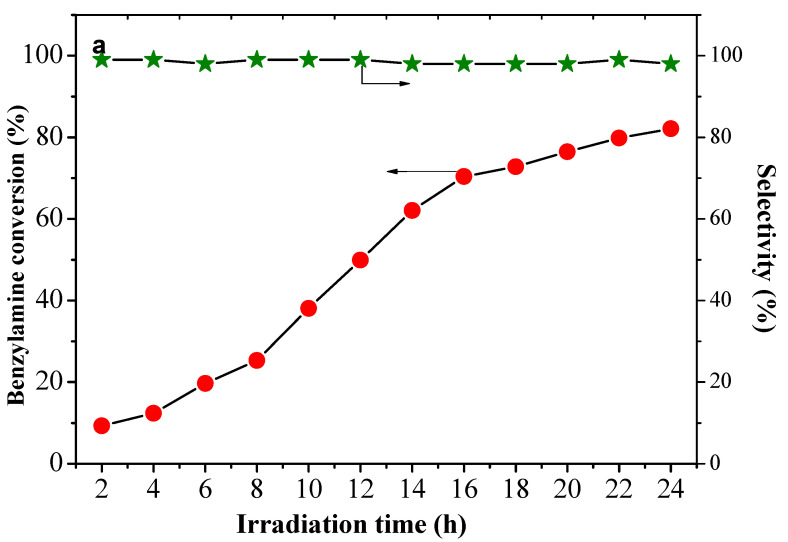
(**a**) Influence of irradiation time on conversion of benzylamine and selectivity of corresponding imine with CuW_0.7_Mo_0.3_O_4_ (x = 0.7); (**b**) yields of CH_3_OH production as a function of irradiation time.

**Figure 7 nanomaterials-10-01303-f007:**
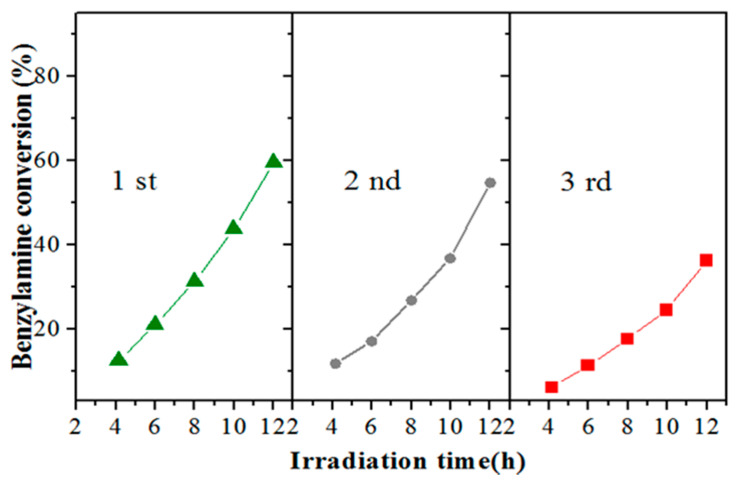
Photocatalytic activity of the reuse of CuW_0.7_Mo_0.3_O_4_ (x = 0.7).

**Table 1 nanomaterials-10-01303-t001:** Molar ratio of Cu:Mo:W.

Entry	The Value of x	The Ratio of Cu:Mo:W
1	0.5	1:0.12:0.97
2	0.7	1:0.23:1.05
3	0.8	1:0.76:0.13

**Table 2 nanomaterials-10-01303-t002:** Photocatalytic activity of different nanoparticles.

Entry	Catalyst	Gas	CH_3_OH	Amine	Imine
	the Value of x		(μmol/g)	Conv. (%)	Sel. (%)
1	0	CO_2_	296.9	29.5	>99
2	0.3	CO_2_	375.4	35.9	>99
3	0.5	CO_2_	571.2	48.3	>99
4	0.7	CO_2_	671.8	54.2	>99
5	0.8	CO_2_	509.9	45.7	>97
6	0.9	CO_2_	394.6	34.4	>99
7	1.0	CO_2_	249.3	23.1	>99
8 ^a^	0.7	N_2_	--	2.1	--
9 ^b^	--	CO_2_	--	1.7	--
10 ^c^	0.7	CO_2_	51.7	9.5	>99
11 ^d^	0.7	CO_2_	--	--	--
12 ^e^	0.7	CO_2_	--	91.3	--
13 ^f^	--	CO_2_	--	7.1	--

^a^ 1 mmol benzylamine; 40 mL CH_3_CN, 60 mg photocatalyst; ^b^ without photocatalyst; ^c^ 40 mL benzylamine, without CH_3_CN; ^d^ 40 mL CH_3_CN, without benzylamine; ^e^ O_2_, ^f^ O_2_, without photocatalyst.

**Table 3 nanomaterials-10-01303-t003:** ^a^ Effect of different amines on photocatalytic activity.

Entry	Reactant	Product	Yield ^b^	Conv. (%) ^c^	Sel. (%) ^d^
1	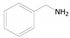	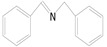	671.8	57.2	99
2	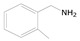	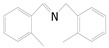	127.5	16.3	99
3	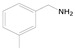	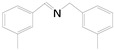	227.4	26.2	98
4	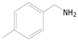	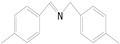	251.9	43.9	99
5	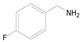	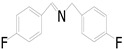	33.1	6.5	98
6	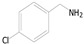	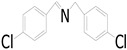	137.2	19.4	98
7	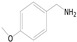	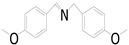	214.6	25.1	99
8	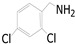	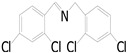	221.4	30.6	99
9	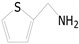	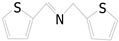	126.2	16.3	98
10	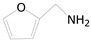	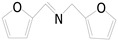	29.7	4.9	98
11	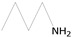	--	--	--	--
12	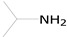	--	--	--	--

^a^ 1 mmol amine, 0.06 g photocatalysis, 40 mL CH_3_CN, 0.1 Mpa CO_2_; ^b^ Unit: (µmol g^−1^); ^c^ Conversion of amine; ^d^ Selectivity of the corresponding imine.

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
