# Peer review of "CuMoxW(1-x)O4 Solid Solution Display Visible Light Photoreduction of CO2 to CH3OH Coupling with Oxidation of Amine to Imine"

_nanomaterials, 2020, doi:10.3390/nano10071303_

Round 1
Reviewer 1 Report
In this articles, the authors fabricated CuMoxW(1-x)O4 solid solution photocatalyst and achieved CO2 reduction under visible light. In addition, by combining amine oxidation, imine as well as methanol were obtained simultaneously. Through the systematic study, the authors found the optimum ratio of the photocatalyst.
Because the content is scientifically correct and interesting, I recommend to publish this article in Nanomaterials after some minor revisions.
Here lists the comments.
1. The authors claimed that they did not use scarified regents in Line 54. However, amine was used to capture the photo-generated holes. In my feeling, “scarified regents” is a chemical to capture the active species (including holes) formed by photocatalytic reaction. Thus, I am confusing.
2. SEM image in Figure 3(a) is too dark, so, please adjust the contrast to improve the visibility.
3. In line 211, the authors wrote that there are slight differences in the conversion process between o-, m-, and p- methyl benzylamines. However, from Table 3, I feel the yield of methanol and conversion efficiency of amine differs among o-, m-, and p- positions.
4. In line 211-218, the authors mentioned that electron donating substituents promoted amine conversion than electron withdrawing substituents. In addition, they mentioned that heterocyclic amines could be also converted to imines, while aliphatic amines could not. Because the reason of these is very important in this study, I strongly recommend to include in the revised manuscript.
5. There are several simple mistakes. To give an example, “1. Experimental” in Line 58 should be “2. Experimental”. The font used in “nuclear magnetic resonance” in Line 62 is different. Figure 6(b) is missing in Figure 6 (i.e., two figures are same). Please correct them and carefully check in order not to leave any mistakes during the revision.
Author Response
Replies to reviewer 1
First of all, thank you for reviewing our paper. Your comments are very important. We have revised the manuscript carefully according to your following comments and suggestions. The detailed replies are as follows.
In this articles, the authors fabricated CuMoxW(1-x)O4 solid solution photocatalyst and achieved CO2 reduction under visible light. In addition, by combining amine oxidation, imine as well as methanol were obtained simultaneously. Through the systematic study, the authors found the optimum ratio of the photocatalyst. Because the content is scientifically correct and interesting, I recommend to publish this article in Nanomaterials after some minor revisions.
Reply: Thank you very much for your comments. We have revised the manuscript carefully according to your following comments and suggestions. The detailed replies are as follows.
1.The authors claimed that they did not use scarified regents in Line 54. However, amine was used to capture the photo-generated holes. In my feeling, “scarified regents” is a chemical to capture the active species (including holes) formed by photocatalytic reaction. Thus, I am confusing.
Answer:Thank you very much for your comments. In fact, amines not only act as sacrificial agents but also are oxidized to imines, which is the important industrial chemicals. So we think it not act as scarified regents and the whole process is economic.
2.SEM image in Figure 3(a) is too dark, so, please adjust the contrast to improve the visibility.
Answer: According to your suggestion, we have adjusted the contrast of the SEM image in Figure 3(a). The revised images are shown as below.
Fig. 3 (a) SEM (b) TEM, and (c), (d) HR-TEM images of the CuMo0.7W0.3O4 (x = 0.7) photocatalyst.
3.In line 211, the authors wrote that there are slight differences in the conversion process between o-, m-, and p- methyl benzylamines. However, from Table 3, I feel the yield of methanol and conversion efficiency of amine differs among o-, m-, and p- positions.
Answer: For this, it can be seen from the Table 3 that the yield of methanol and conversion efficiency of amine in the process turn out to be very different among o-, m-, and p- methyl-substituted benzylamines. For clearly, we have already corrected the sentence “but there are slight differences in the conversion process” to ‘but there are slight differences in the conversion process and it can been seen from the Table 3 that their conversion rates differ vary greatly, which might to be relative to the effect of the steric hindrance’ . The whole corrected sentence has marked in the red color in the revised manuscript.
4.In line 211-218, the authors mentioned that electron donating substituents promoted amine conversion than electron withdrawing substituents. In addition, they mentioned that heterocyclic amines could be also converted to imines, while aliphatic amines could not. Because the reason of these is very important in this study, I strongly recommend to include in the revised manuscript.
Answer:Thank you very much for your comments. Indeed as is mentioned that heterocyclic amines could be also converted to imines, while aliphatic amines could not. Compared with heterocyclic imines, aliphatic imines are extremely unstable, which leads to the the reaction(RCH2NH2 + 2p+val → RCH=NH + 2H+cond ) of aliphatic amines to imines reversible. So aliphatic amines could not be converted to imines.
5.There are several simple mistakes. To give an example, “1. Experimental” in Line 58 should be “2. Experimental”. The font used in “nuclear magnetic resonance” in Line 62 is different. Figure 6(b) is missing in Figure 6 (i.e., two figures are same). Please correct them and carefully check in order not to leave any mistakes during the revision.
Answer:Thank you very much for your advise. I am so sorry for my mistakes. I have corrected it in the manuscript and marked in red. In the end I have carefully reviewed the full text and guarantee no similar mistake.

Reviewer 2 Report
In this paper, Wennan Zou and Hongming Wang describe the use of CuMoxW(1-x)O4 solid solution for the Photoreduction of CO2 to CH3OH with the oxidation of benzylamines to imines. The paper is well written, and is interesting for the development of sustainable catalytic methodologies. However, in order to be accepted to Nanomaterials major revisions should be done:
- The authors should provide the supplementary material. I could not revise these material.
- Line 144: Cu3(MoO4)2(OH)2 is in another fond of the rest of the manuscript
- In Table 2, there are several mistakes, in entry 11, why there is selectivity if there are no benzylamine. Entry 12, the selectivity is missing.
- In the tables the hours of irradiation, wavelength and lamp should be indicated.
- In the explanation in line 184, indicate that the reactions are with O2.
- Figure 6a and 6b, are the same. Please provide the figure with the conversion of MeOH. Please indicate what is the green star and the red circle.
- Why benzylamine is the best amine. The authors should explain why electron-withdrawing groups (F, Cl) and electron-donating (Me, MeO) gave lower yields for MeOH production.
- Table 3, the authors should improve the drawing of the amines are imines, it should be homogeneous.
Author Response
Replies to reviewer 2
First of all, thank you for reviewing our paper. Your comments are very important. We have revised the manuscript carefully according to your following comments and suggestions. The detailed replies are as follows.
In this paper, Wennan Zou and Hongming Wang describe the use of CuMoxW(1-x)O4 solid solution for the Photoreduction of CO2 to CH3OH with the oxidation of benzylamines to imines. The paper is well written, and is interesting for the development of sustainable catalytic methodologies. However, in order to be accepted to Nanomaterials major revisions should be done:
Reply: Thank you very much for your comments. We have revised the manuscript carefully according to your following comments and suggestions. The detailed replies are as follows.
1.The authors should provide the supplementary material. I could not revise these material.
Answer: We have submitted the supplementary materials in revised manuscript.
2.In Table 2, there are several mistakes, in entry 11, why there is selectivity if there are no benzylamine. Entry 12, the selectivity is missing.
3.In the tables the hours of irradiation, wavelength and lamp should be indicated.
Answer:We are sorry for the incorrect data, we have corrected Table 2 carefully. This series of comparative experiments in Table 2 are designed to illustrate the optimization of catalysts and their catalytic conditions, and the results suggest that the .CuW0.7Mo0.3O4 (x = 0.7) solid solutionis is the best catalyst after 12 hours irradiation , which has the highest yields of methanol and imine. The revised Table 2 is shown as follows.
4.In the explanation in line 184, indicate that the reactions are with O2.
Answer: Because the reaction takes place in an airtight container, no oxygen is involved.
5.Figure 6a and 6b, are the same. Please provide the figure with the conversion of MeOH. Please indicate what is the green star and the red circle.
Answer: In Figure 6a, green star represents the selectivity of imine and red circle represents the conversion of benzylamine. And yields of CH3OH production as a function of irradiation time are illustrated in Figure 6b.
Fig. 6 (a) Influence of irradiation time on conversion of benzylamine and selectivity of corresponding imine with CuW0.7Mo0.3O4 (x = 0.7)ï¼›(b) yields of CH3OH production as a function of irradiation time.
6.Why benzylamine is the best amine. The authors should explain why electron-withdrawing groups (F, Cl) and electron-donating (Me, MeO) gave lower yields for MeOH production.
Answer: The conclusion that benzylamine is the best amine may be due to its high activity under the photocatalyst condition。
7.Table 3, the authors should improve the drawing of the amines are imines, it should be homogeneous.
Answer:Thank you very much for your suggestion,we have modificated it.The corrected table is revised as follows.

Round 2
Reviewer 2 Report
this manuscript can be accepted for publication in Nanomaterials